# Teaching Uncertainty Quantification in Machine Learning through Use Cases

**Matias Valdenegro-Toro** [1]

## Abstract

Uncertainty in machine learning is not generally taught as general knowledge in Machine Learning course curricula. In this paper we propose a short curriculum for a course about uncertainty in machine learning, and complement the course with a selection of use cases, aimed to trigger discussion and let students play with the concepts of uncertainty in a programming setting. Our use cases cover the concept of output uncertainty, Bayesian neural networks and weight distributions, sources of uncertainty, and out of distribution detection. We expect that this curriculum and set of use cases motivates the community to adopt these important concepts into courses for safety in AI.

## 1. Introduction

Neural networks and machine learning models are ubiquitous in real-world applications, but in general model and data uncertainty are not well explored, and this propagates on how machine learning is taught at different levels. Uncertainty is an important concept that should be taught to all students interested in machine learning.

Overall Uncertainty Quantification of machine learning models (Gawlikowski et al., 2021) is not part of the standard curricula at the undergraduate or graduate level, mostly being present in advanced summer schools (like MLSS, EEML, DeepLearn, SMILES, etc), with some exceptions at graduate courses aimed mostly at theory of Bayesian NNs (BNNs).

In this paper we aim to develop a concept for teaching uncertainty quantification in machine learning, first with a short curriculum, and then through different use cases, starting from why we need models with uncertainty and ending at out of distribution detection. We hope that this material can be used for easier planning of future courses.

[1] German Research Center for Artificial Intelligence, Bremen, Germany. Correspondence to: Matias Valdenegro-Toro <matias.valdenegro@dfki.de>.

*Proceedings of the $2^{nd}$ Teaching in Machine Learning Workshop*, PMLR, 2021. Copyright 2021 by the author(s).

Teaching with clear use cases can be beneficial for student's learning (Lynn Jr, 1999), specially when they are combined with practical experience.

Uncertainty in ML is a subject that is heavy on probability and statistics, and this is a topic that might not be easy for some students. We believe that having clear use cases for this purpose can help students learn and to clarify concepts. These use cases can be implemented in code using standard machine learning frameworks like Keras, TensorFlow, and PyTorch.

## 2. Curricula for UQ in ML

We first introduce a short curricula template for a uncertainty in machine learning course. This could be a graduate-level course, requiring students to know basic neural networks, machine learning theory, and probability and statistics, as well as having appropriate coding skills in a programming language in order to understand and implement the use cases in a framework of their choice.

The overall curriculum is presented in Table 1. Any teacher should of course adapt this course to their institution or student body, and we encourage the teacher to also include seminar-style discussions including state of the art research in BNNs and uncertainty in ML, as this is still a very research heavy field.

The ultimate goal of this course is to enable students to perform research in this field, and to apply this knowledge into neighboring task field like Computer Vision, Reinforcement Learning, or Robotics.

## 3. Use Cases

In this section we present a selection of use cases to teach concepts of uncertainty in machine learning settings. These represent what we think are the most difficult concepts for students to grasp, which motivate the application of use cases as teaching methodology.

### 3.1. Output Uncertainty

The best use case to teach the concept of uncertainty at the output of a machine learning model is a simple regression setting, as the output mean can be associated to the output

| Unit | Content |
|------|---------|
| Introduction to UQ | Point-wise outputs versus distribution outputs in ML models. Two-headed models for regression. Sources of uncertainty. Representations of output uncertainty. Applications and possible legal requirements. Relationship to Explainable AI. Connections to safety and trustworthiness in AI. |
| Statistical Methods | Categorical, Gaussian, and Dirichlet Distributions. Predictive intervals, Quantile Regression. |
| Bayesian NNs | Distribution over weights. Predictive posterior distribution. Inference using Bayes Rule. |
| Methods for UQ | Deep Ensembles (Lakshminarayanan et al., 2016), Monte Carlo methods like Dropout (Gal & Ghahramani, 2016) and DropConnect (Mobiny et al., 2019). For advanced courses, Gaussian Processes (Rasmussen & Williams, 2005) and Markov Chain Monte Carlo methods (Betancourt, 2017) can also be included. |
| Metrics and Evaluation | Losses with uncertainty, Entropy, Calibration, Reliability plots, and related calibration metrics (Guo et al., 2017). Advanced topics can be proper scoring rules (DeGroot & Fienberg, 1983). |
| Out of Distribution Detection | In distribution and Out of distribution data. Evaluation protocol with standard datasets (CIFAR10 vs SVHN, MNIST vs Fashion MNIST). Evaluation using histograms and ROC curves. |
| Challenges and Future Research | Scalability of BNNs, Generalization of out of distribution detection, Computational performance, Datasets with uncertainty, and Real-world applications (Valdenegro-Toro, 2021). |

*Table 1.* Curriculum for a graduate course in Uncertainty Quantification in Machine Learning

of a classical model (without uncertainty), and the standard deviation of the output can be directly associated with the uncertainty in the output. In a classification setting with probabilities associated to each class, it is more difficult to directly see the effect of uncertainty in the model.

**Learning Objective**. Students will learn about the difference between a classical machine learning model and one with output uncertainty.

**Use Case**. Students will implement a standard neural network using a framework of their or the teacher's choice. Students will generate data by sampling the following function:

$$f(x) = \sin(x) + \epsilon \tag{1}$$
$$\epsilon \sim \mathcal{N}(0, \sigma(x)) \tag{2}$$
$$\sigma(x) = 0.15(1 + e^{-x})^{-1} \tag{3}$$

For the range $x \in [-\pi, \pi]$. Two neural network models can be used. One is a standard neural network and the other is a ensemble of 5 neural networks (Lakshminarayanan et al., 2016), which is a simple method to estimate uncertainty. An example of this setting can be seen in Figure 1, where output uncertainty is represented as confidence intervals. Students can the visually compare their results, and relate on how the standard neural network does not model the training data points variations, while the neural network with uncertainty does. This is specially noticeable as the standard deviation of the noise is variable, which is not captured with a standard neural network.

A variation of this exercise is to use a deep ensemble, where each ensemble member has two output heads, one for the task ($\mu(x)$) and another for uncertainty ($\sigma^2(x)$), which can be trained with a negative log-likelihood loss that does not require labels for uncertainty (Eq 5). This exercise helps students see that a model needs to be "added something" to estimate uncertainty properly, an output head in this case. The uncertainty head $\sigma^2(x)$ represents the variance of the output .

### 3.2. Bayesian Neural Networks

Bayesian Neural Networks are difficult to understand conceptually since the formulation is heavy in probability, and weights are replaced by probability distributions. In this use case we simplify the concept for easy understanding.

**Learning Objective**. Students will learn the conceptual differences between a standard and an Bayesian neural network and how it relates to produce uncertainty at the model output.

**Use Case**. Students will implement the forward pass of a simple neural network using numpy or a similar linear algebra framework. For a standard neural network, scalar or point-wise weights are used, and for a BNN, weights will be drawn from a given Gaussian distribution (the actual weight values for this use case do not matter). Sampling can be used to produce predictions from a BNN, by sampling a set of weights and producing a forward pass with a given input.

Students will compare the outputs given random weights for each of their networks, and compare how the BNN is a stochastic model, meaning that predictions vary with a given input, as different weights are sampled and propagate

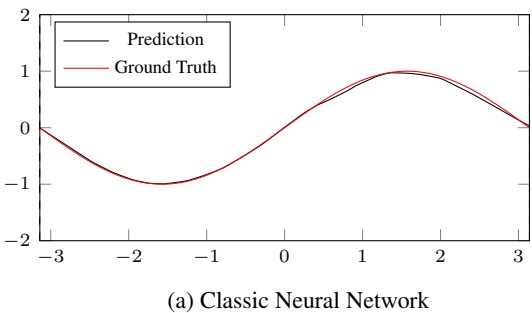 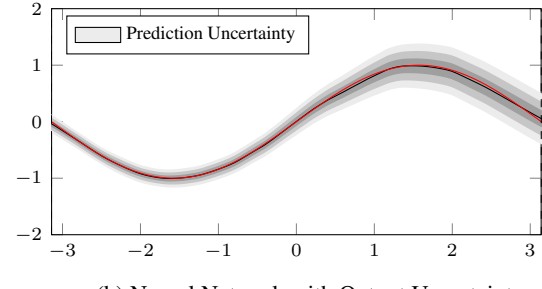

(a) Classic Neural Network          (b) Neural Network with Output Uncertainty

*Figure 1.* Comparison between classic and neural networks with output uncertainty for regression of $f(x) = \sin(x) + \epsilon$.

through the network to produce different outputs, but these predictions are not completely random, and are samples of the predictive posterior distribution.

In comparison, the standard neural network has fixed predictions with a given input and weights, which cannot model uncertainty. An additional experiment is to vary the depth or width of the network, as a way to increase the number of weights, and see how predictions change in terms of stochasticity.

### 3.3. Bayesian NN Intractability

Connecting to the previous use case, it is well known that inference in BNNs is intractable, due to the high computational complexity required to estimate weight distributions, particularly for highly parameterized neural networks. In this use case we wish the student to form an intuition on why this is the case.

**Learning Objective**. Students will learn an intuition on why BNNs are intractable with a thought experiment and validate it with a code implementation.

**Use Case**. As a thought experiment, students should think about the predictive posterior distribution (Eq 4), which integrates a term over the weights of the network to produce a distribution output.

$$P(y \,|\, x) = \int_w P(y \,|\, x, w) P(w) dw \qquad (4)$$

For the experimental setting, students should implement a simple BNN using numpy, with randomly initialized weight distributions (Gaussian distributions can be used for simplicity), and then produce predictions with random data (similarly to the previous use case). But then students are asked to vary their network architectures, increasing depth from a few layer to over 50 layers, or the width from a small number to a large number (over 1024 neurons), and then estimate and plot the computation time as network depth and number of samples is varied.

Students the analyze their results and comment on the appli-

cability of BNNs for real-world applications, considering their computational costs. Additional experiments for discussion are the possibilities of computing the integral in Eq 4 with analytical or numerical methods.

### 3.4. Aleatoric vs Epistemic Uncertainty

Different sources of uncertainty (Der Kiureghian & Ditlevsen, 2009) and their separation (Kendall & Gal, 2017) are not always easy to see and learn intuitively. This use case tries to show the difference with a practical example in a regression setting.

**Learning Objective**. Students will learn the difference between aleatoric and epistemic uncertainty through a simple regression problem, and how different parts of the model contribute to these sources of uncertainty.

**Use Case**. We will use the same setting as the output uncertainty use case (Sec 3.1), but only a model with uncertainty. Since an ensemble is used to estimate uncertainty in this case, we will use the negative log-likelihood loss formulation to estimate aleatoric uncertainty:

$$L(y_n, \mathbf{x}_n) = \frac{\log \sigma_i^2(\mathbf{x}_n)}{2} + \frac{(\mu_i(\mathbf{x}_n) - y_n)^2}{2\sigma_i^2(\mathbf{x}_n)} \qquad (5)$$

Students should train an ensemble of 5 networks, and then plot the predictions separately. First students plot the predictions of the mean output of each ensemble member (which produces epistemic uncertainty), and then separately plot the standard deviation outputs of each ensemble member (which estimate aleatoric uncertainty). This concept is shown in Figure 2.

Students then compare both kinds of predictions and try to explain the differences, and how they relate to the epistemic and aleatoric sources of uncertainty. A plot of the training data might also help students visualize aleatoric uncertainty.

### 3.5. Out of Distribution Detection

Out of distribution detection entails detecting input samples outside of the training set distribution, through output un-

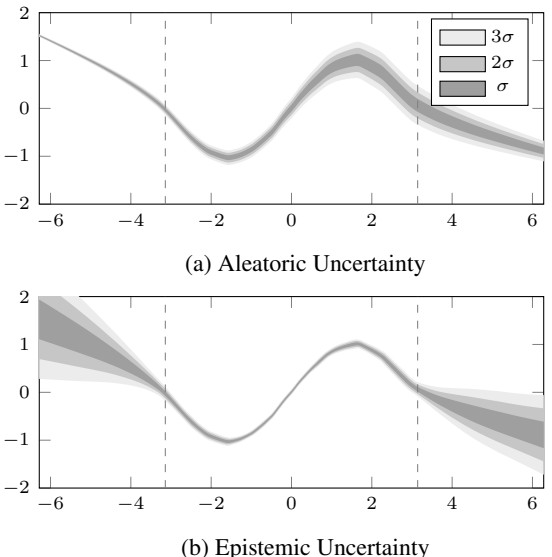

(a) Aleatoric Uncertainty

(b) Epistemic Uncertainty

*Figure 2.* Comparison between Epistemic and Aleatoric Uncertainty in the Toy Regression example.

certainty or other confidence measures. In this setting we present two use cases.

**Learning Objective**. Students will learn how to perform and evaluate out of distribution using standard image classification datasets and in a regression toy example, and to get the intuitions on how uncertainty enables the out of distribution detection task.

**Classification Use Case**. Using an appropriate neural network framework, students will implement and train a BNN (or an approximation) on the SVHN dataset (ID, in-distribution), and evaluate in the train and test splits. Then students are asked to make predictions using their model on the CIFAR10 test set (OOD, out-of-distribution) and to look at the class probabilities that their model produces. Entropy can be used to obtain a single measure for each sample, and then compare the ID vs OOD entropy values using a histogram. The use case can be completed by obtaining a threshold between ID and OOD entropy distributions using an ROC curve, in order to perform out of distribution detection in the wild.

**Regression Use Case**. For a toy example in regression, we use the same setting as Sec 3.1, keeping the training set $x \in [-\pi, \pi]$, and introducing an OOD set of $x \in [-2\pi, -\pi] \cup [\pi, 2\pi]$. Students then plot the predictions from their model, noting the values on the two datasets (ID and OOD). A sample result can be seen in Figure 3.

Students should compare the output standard deviation produced by their model in the ID and OOD datasets. They should observe that uncertainty as predicted by the output standard deviation should be higher in the OOD data than in

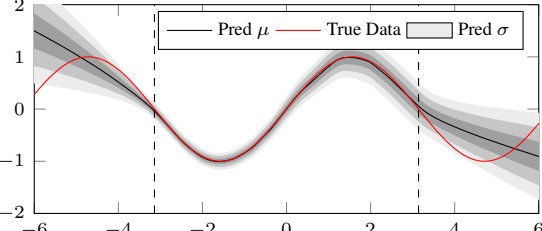

*Figure 3.* Out of Distribution Detection in the Toy Regression Example. Values $x > \pi$ and $x < -\pi$ are out of distribution in this example, which triggers large epistemic uncertainty and can be used to detect this condition.

the ID data, which indicates that the model is extrapolating. Students can add additional evidence of extrapolation by plotting the function $f(x) = \sin(x)$ which is the true function that generated the training data, and confirm that the model predictions in the OOD data are very incorrect when compared to the true function, while predictions in the ID data are correct inside the training range ($[-\pi, \pi]$). Error metrics like mean absolute error can be used to confirm this difference.

The teacher can also show that uncertainty in the OOD set should be proportional to the distance (in input space) from the sample to the edge of the OOD set, and that this proportionality is expected for proper uncertainty quantification.

**Misconceptions**. Students might be confused that some OOD examples have low uncertainty and are easily confused with ID examples. This can be explained with models are not perfect and make mistakes, and this also translates into mistakes in OOD.

Another issue is the definition of out of distribution data can be very abstract, as it is an open set that corresponds to anything not in the training data distribution. Multiple OOD datasets can be used to show this.

## 4. Conclusions and Future Work

In this paper we have presented a small course curriculum and a selection of use cases to teach students about uncertainty quantification in machine learning models. We hope that this work can motivate the community about the importance of teaching uncertainty quantification and BNNs to students learning about machine learning, and how it relates to the concept of safety in artificial intelligence.

Future course contents and use cases can be centered in specific applications of machine learning and artificial intelligence, such as Computer Vision, Robotics, or Autonomous Systems. There is a good demand to connect theoretical fields (BNNs in particular) into practical applications as a way to lead future research.

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
