# OpenReview forum: "Teaching Uncertainty Quantification in Machine Learning through Use Cases"
_ecmlpkdd.org/ECMLPKDD/2021/Workshop/TeachML — TeachML 2021_

### Official Review · Reviewer_qczQ · 2021-07-04
**Use Cases to help teach uncertainty quantification**

**Rating:** 8
**Confidence:** 4

**Review:**

The paper is proposing a set of use cases to be proposed to ML students to help them grasp a number of issues about uncertainties in ML. The paper is well written with a very adequate bibliography. It covers Bayesian NN but also classical NN.
Indeed uncertainty is often not well covered in ML courses, while it is quite important in any applications. The paper suggests first a set of topics to be covered in the course, before proceeding to detail use cases to be covered in practical works. A first use case is to use an ensemble of NN for a regression task, with a simple ensemble of NN to evaluate the uncertainty. The case of a classifier trained with N-fold cross validation would also have been suitable. The second and third use case are random Bayesian NN, however no attempt is made to train a BNN, probably because of the computation cost; I believe having the students loaded a precomputed BNN would be more informative. The fourth cases is the study in the same regression case of aleatoric vs epistemic uncertainty. The fifth use cases is about the very important point of Out of Distribution Detection, for regression and a classification task.

Overall the paper proposes a suitable set of use cases to give ML students good intuition about uncertainty in ML. I would have wished: 1) to see example code for the different use cases 2) to see N fold cross validation 3) deeper use of BNN, alleviating the large training resources with pre-computed models to be loaded.

---

### Official Review · Reviewer_LM1U · 2021-07-08
**A detailed course outline for a so far under-represented topic in machine learning**

**Rating:** 8
**Confidence:** 3

**Review:**

The authors present an approach for creating a course on uncertainty quantification of machine learning applications.
I believe this is an important topic but often overlooked and I find it great to see that the authors put in the effort to create a course outline for this topic!
I wished the following information was contained in the material:
    - whether the course material exists already
    - whether it has been taught already
    - what the time frame of that course is

The structure of the presented course is thoughtfully designed, going from the basics to an application of uncertainty quantification step-by-step, as well as providing learning objectives for each course unit.
Also, the focus on clear and relatively simple use cases might make understanding of concepts easier. However, (depending on the time frame) I believe the stated goal of the course (to be able to conduct research in this field) may be a bit too far fetched.

Below are my comments regarding the single units of the proposed course:

For unit 1, I feel there is a part missing explaining that a network with built-in uncertainty needs to have two outputs, one for the task and one for the uncertainty. The term "output mean" comes a bit unexpected and has not been introduced before.

Figure 1a): What are the red line and black line respectively? A legend would help.
Figure 1b): please specify in the caption that the uncertainty comes from the standard deviation of the 5 models that were used as an ensemble. Maybe 5 is also a bit low for computing a standard deviation and the ensemble size should be increased.

For unit 2 I especially like the design of introducing bayesian neural networks by manually coding them and believe that this will speed up understanding of the concept.

Regarding unit 3 I am a bit unsure. Although I think emphasizing the intractability of inference is important, I'm not sure it is enough to demonstrate that computation times are increasing with larger models (since students could think that by scaling out to better hardware the problem would be solved easily)

In unit 4 the different types of uncertainty (aleatoric, epistemic) should be introduced but the paper did not contain a clear explanation of those uncertainty sources.
There should definitely also be a motivation or derivation of the used likelihood function.

Figure 2: the picture to me suggests that the difference between aleatoric and epistemic uncertainty is to be uncertain about in-sample and out-of-sample points but I'm not sure if that is capturing the full meaning of those terms correctly.

Unit 5 is missing the last part about the misconceptions and the explanation of the authors on how to avoid confusion would be really valuable!

---

### Decision · Program_Chairs · 2021-07-21

**Decision:**

Accept

**Comment:**

Congratulations! The reviewers agree that this paper should be accepted.

Camera-ready version is due August 18, 2021. As you prepare the camera ready version, please take the reviewers comments into consideration.

We look forward to your participation at the workshop on September 13, 2021. We invite you also to join us for the satellite event on September 08, 2021. Schedules for both the workshop and the satellite event will be forthcoming.